

# Mineralogical properties and internal structures of individual fine particles of Saharan dust

Gi Young Jeong[*,1], Mi Yeon Park[1], Konrad Kandler[2], Timo Nousiainen[3], Osku Kemppinen[3,4]

[1]Department of Earth and Environmental Sciences, Andong National University, Andong 36729, Republic of Korea
[2]Institut für Angewandte Geowissenschaften, Technische Universität Darmstadt, Schnittspahnstr. 9, 64287 Darmstadt, Germany
[3]Earth Observation, Finnish Meteorological Institute, P.O. Box 503, FI-00101, Helsinki, Finland
[4]Department of Applied Physics, Aalto University, Espoo, P.O. Box 11000, FI-00076, Aalto, Finland

[*]*Correspondence to*: G.Y. Jeong (jearth@anu.ac.kr)

**Abstract**: Mineral dust interacts with incoming/outgoing radiation, gases, other aerosols, and clouds. The assessment of its optical and chemical impacts requires knowledge of the physical and chemical properties of bulk dust and single particles. Despite the existence of a large body of data from field measurements and laboratory analyses, the internal properties of single dust particles have not been defined precisely. Here, we report on the mineralogical organization and internal structures of individual fine (< 5 μm) Saharan dust particles sampled at Tenerife, Canary Islands. The bulk of Tenerife dust was composed of clay minerals (81%), followed by quartz (10%), plagioclase (3%), and K-feldspar (2%). Cross-sectional slices of Saharan dust particles prepared by the focused ion beam technique were analyzed by transmission electron microscopy (TEM) to probe the particle interiors. TEM analysis showed that the most common particle type was clay-rich agglomerate, dominated by illite–smectite series clay minerals with subordinate kaolinite. Submicron grains of iron (hydr)oxides (goethite and hematite) were commonly dispersed through the clay-rich particles. The median total volume of the iron (hydr)oxide grains included in the dust particles was about 1.5 vol% . The average iron content of clay minerals, assuming 14 wt% $H_2O$, was determined to be 5.0 wt%. Coarse mineral cores, several micrometers in size, were coated with thin layers of clay-rich agglomerate. Overall, the dust particles were roughly ellipsoidal, with an average axial ratio of 1.4:1.0:0.5. The mineralogical and structural properties of single Saharan dust particles provide a basis for the modeling of dust radiative properties. Major constituent minerals, such as illite–smectite series clay minerals and iron (hydr)oxides, were commonly submicron- to nano-



sized, possibly enhancing their biogeochemical availability to remote marine ecosystems lacking micronutrients.

## 1 Introduction

Mineral dust affects Earth's climate and ecosystems via interaction with electromagnetic radiation (Sokolik and Toon, 1996; Tegen and Lacis, 1996; Formenti et al., 2011), acidic gases, and anthropogenic aerosols (Dentener et al., 1996; Ooki and Uematsu, 2005; Laskin et al., 2005), by becoming ice condensation nuclei (Kulkarni and Dobbie, 2010; Freedman, 2015), and delivering micronutrients to remote ocean and terrestrial environments (Swap et al., 1992; Mahowald et al., 2009; Johnson and Meskhidze, 2013). To understand the interactions of dust with environments, a wide range of studies have been carried out to characterize the physical and chemical properties of bulk dust and individual dust particles, such as the size distribution (Seinfeld et al., 2004), particle morphology (Okada et al., 2001; Reid et al., 2003), chemical composition (Coude-Gaussen, 1987; Okada et al., 1990; Kanayama et al., 2002; Ro et al., 2005; Klaver et al., 2011; Arimoto et al., 2004), and mineral composition (Glaccum and Prospero, 1980; Avila et al., 1997; Shi et al., 2005; Kandler et al., 2007, 2009; Jeong, 2008; Jeong et al. 2014).

Of these properties, the internal structure and mineralogical properties of individual dust particles have been the least studied, largely due to the lack of suitable and reliable methods. Scanning electron microscopy (SEM) combined with energy-dispersive X-ray spectrometry (EDXS) has been applied to obtain information on the two-dimensional (2D) morphology and overall chemical composition of individual particles (Okada et al., 1990; Anderson et al., 1996; Ro et al., 2005; Gao et al., 2007). These individual dust particles, however, are usually composed of polymineralic/polycrystalline mixtures of diverse minerals (Falkovich et al., 2001; Jeong, 2008), and SEM-EDXS cannot easily yield information on their internal structures and mineralogical properties. Focused ion beam (FIB) technique was applied to expose the cross sections of urban dust particles for SEM analysis (Conny, 2013). A practical method for the acquisition of data on the internal properties, however, was transmission electron microscopy (TEM) for thin cross-sectional slices prepared from single dust particles by FIB technique. With these improvements, Jeong and Nousiainen (2014) reported internal structures and mineralogical makeup of single particles of Asian dust which could be classified into several structural types.

Knowledge of the internal properties of dust particles is essential for the establishment of a realistic model of



mineral dust, which would enable assessment of its radiative properties and biogeochemical contributions. Based on the findings of Jeong and Nousiainen (2014), Kemppinen et al. (2015) calculated the optical properties of internally inhomogeneous single dust particles, and showed that their light scattering properties depended significantly on their internal structures and iron oxide mineralogy of dust particles, as well as how they were

accounted for in the modeling. The difficulty of accurately mimicking the impacts of these inhomogeneities on scattering by effective medium approximations suggests that accurate optical models of dust particles should account explicitly for at least the most impactful types of internal inhomogeneity. Regarding biogeochemical contributions, mineral dust transported to remote marine ecosystems is reportedly responsible for phytoplankton blooms due to the supply of iron, and terrestrial ecosystems are also considerably impacted (Swap et al., 1992;

Boyd et al., 2000; Jickells et al., 2005; Formenti et al., 2011). The bioavailability of iron in oceans depends largely on its mineralogical form in the dust particles (Cwiertny et al., 2008; Journet et al., 2008), which can also be investigated on FIB slices using TEM.

The Saharan desert is the largest single source of atmospheric dust. Large volumes of this emitted dust are transported over long distances across the Atlantic Ocean and Mediterranean Sea. Although the physical,

chemical, and mineralogical properties of bulk Saharan dust and single particles have been investigated (Falkovich et al., 2001; Reid et al., 2003; Kandler et al., 2007, 2009; Formenti et al., 2011), the mineralogical compositions and internal structures of single dust particles have not been studied. Jeong and Achterberg (2014) attempted TEM analysis of FIB slices prepared from individual Saharan dust particles, but unsuccessfully as original individual dust particles could not be identified on their filter heavily deposited with dust. To extend the

mineralogical and structural analyses of Asian dust particles conducted by Jeong and Nousiainen (2014), we have now conducted a similar analysis of FIB slices prepared from fine Saharan dust particles collected specifically for single-particle electron microscopic analysis.

## 2   Samples and methods

Dust samples were collected from the top of the tower building of the meteorological observatory of Izaña, Tenerife, Spain (28° 18' 33.8" N, 16° 29' 56.9" W, 2395 m a.s.l.). Details on the procedure and location can be found in Kandler et al. (2007). Fine (< 5 µm) particles were collected with a cascade impactor on carbon adhesive



for 10 min on 15 July 2005 at 09:33 h (UTC). All samples were stored in a desiccator under dry conditions prior to analysis.

Backward trajectories of air mass were calculated by HYSPLIT4 rev. 2015-06-16 (Stein et al., 2015). Fig. 1 shows the trajectory plots for the impactor sample. The atmospheric circulation on the sampling day was stable, with a continuous flow leaving the western Saharan desert at altitudes of 3–4 km and travelling over the ocean during the final 2 days before arrival under very dry conditions (10–30% relative humidity). Dust injection is assumed for parts of the trajectory, which were inside the modeled mixing layer. Northeastern Mauritania and southern Algeria are the most probable source regions for the analyzed sample. According to the trajectory analysis, the injected dust had travelled for a minimum of 4 days before it was collected at Izaña. Due to the dry conditions, however, considerable alteration of the dust components was not expected.

Dust particles were analyzed using a TESCAN LYRA3 XMH field emission scanning electron microscope equipped with a Bruker EDXS system. SEM images showed that the individual particles were sufficiently separated from each other, with no coagulation occurring during deposition. EDXS patterns of 1626 particles were obtained and classified according to predominant mineral to obtain approximate mineral compositions of bulk dust (Jeong, 2008; Jeong et al., 2014). The target particles for FIB work were selected according to the mineralogical types of the dust particles, considering their abundance. Thus, the 48 FIB slices analyzed are representative of the Saharan dust particles collected on the sampler. However, three iron-enriched particles were selected for the identification of iron (hydr)oxide mineral species, which were not common.

Dust particles on the carbon adhesive were placed in a JEOL JIB-4601F FIB instrument to prepare thin cross-sectional slices (ca. 6 × 6 μm, <100 nm thickness). Detailed procedures of slice preparation using FIB techniques and a discussion of possible artifacts were provided in Jeong and Nousiainen (2014). The slices are cross sections along the shorter axes of the particles observed by SEM. A total of 48 FIB slices were prepared from 48 dust particles and analyzed using a JEOL JEM 3010 transmission electron microscope for imaging and a JEOL JEM 2010 transmission electron microscope equipped with an Oxford X-MAX EDXS system. TEM images of 18 slices are presented here, and those of 30 slices are presented in Supplementary Fig. 1.

Minerals in the FIB slices were identified by the combined use of EDXS, electron diffraction, and lattice fringe imaging. The identification criteria for clay minerals are summarized in Jeong and Nousiainen (2014). Nano-thin platelets of illite–smectite series clay minerals (ISCMs) have been found commonly in TEM analyses



of natural mineral dusts (Jeong et al., 2014; Jeong and Nousiainen, 2014; Jeong and Achterberg, 2014). They are nano-scale mixtures of nano-thin platelets of illite, smectite, and illite–smectite mixed layer clay minerals, all of which have lattice fringes measuring ~1.0 nm due to the dehydration of smectite interlayer water. Despite the differences in their chemical compositions, the routine identification of these minerals using TEM-EDXS is

challenging because the electron beam diameter could not be reduced below ~100 nm without severe beam damage and count loss. Thus, to avoid over-interpretation of TEM data, nano-thin platelets of clay minerals showing ~1-nm lattice fringes with varying contents of interlayer K and Ca were grouped as ISCMs (Jeong and Nousiainen, 2014). In some cases, when the separate chemical identification of illite and smectite platelets was possible due to the large grain size, we used the terms illite and smectite.

10        The mineralogical identification of iron (hydr)oxide polymorphs was performed using electron diffraction and lattice fringe imaging. The routine unambiguous identification of all iron (hydr)oxide grains, however, was challenging due to the superimposition of d-spacings, varying crystallographic orientations, and often vague electron diffraction patterns. Thus, we used mineralogical names only in cases in which mineral species were identified unambiguously by lattice fringe and electron diffraction; in other cases, we used the collective term

"iron (hydr)oxide". Chemical compositions of ISCMs, illite, smectite, chlorite, and biotite were quantified by measuring X-ray intensities of Si, Al, Fe, Mg, Ti, K, Na, and Ca. The X-ray counts of the elements were converted to weight% using $k$ factors determined experimentally from FIB slices of biotite and plagioclase of known composition from the Palgongsan granite (Jeong, 2000). More details can be found in Jeong and Achterberg (2014), except that a more sensitive Oxford X-MAX EDXS detector was used in this study. In this

study, the term "particle" refers to an individual airborne solid object and the term "grain" refers to a particle constituent.

## 3   Results and Discussion

### 3.1   Mineral composition of bulk dust

Analyses of single-particle SEM-EDXS data showed that ISCMs were the dominant mineral group (69%), followed by kaolinite (11%), quartz (10%), plagioclase (3%), K-feldspar (2%), calcite (1%), chlorite (1%), iron



(hydr)oxide (1%), titanium oxide (1%), and gypsum (1%) (Table 1). The total clay mineral content was 81%. Although we could not obtain X-ray diffraction (XRD) data for bulk Tenerife dust, these single-particle SEM-EDXS data were well matched with XRD data obtained for Saharan dust sampled in Cape Verde (Table 2 in Jeong and Achterberg, 2014). We assume that the Cape Verde sample has similar source regions, as transport of

dust from Mauritania, Mali, and Algeria to Cape Verde in winter is very common (Chiapello et al., 1997; Knippertz et al., 2011). The total clay content determined by XRD for the Cape Verde dust sample was 81%, matching the proportion obtained using single-particle SEM-EDXS data in this study. Scheuvens et al. (2013) proposed, among others, a Ca/Fe ratio for source discrimination of Saharan dust, with very low Ca/Fe ratios for the probable regions. The very low calcite content of the Tenerife Saharan dust sample is thus in agreement with

the identification of source regions by backward trajectory analysis.

## 3.2   Internal structures of individual dust particles

### 3.2.1   Clay-rich particles

Clay-rich particles were the most common (Fig. 2). They were dominated by ISCMs, which were loose, sub-parallel agglomerates of slightly curved nano-thin platelets with 1.0-nm lattice fringes (Fig. 2a). The morphological features of ISCMs contrasted with those of illite which were thick, straight, and compact (Fig. 2a). Very few clay-rich particles were pure ISCMs (Fig. 2b) or kaolinite (Fig. 2c). They were commonly mixtures

with dominant ISCMs and subordinate kaolinite

### 3.2.2   Large minerals with clay-rich coatings

Large mineral grains, such as quartz, plagioclase, and K-feldspar, were usually coated with thin clay layers

composed of ISCMs and kaolinite. The large core grains were roughly equidimensional, with irregular surfaces (Fig. 3). The thicknesses of the clay coatings varied due to the irregular surface topographies of the substrates. Our observations confirmed a near absence of clean surfaces free of clay coatings on quartz, plagioclase, and K-feldspar. Large phyllosilicate grains, such as illite, biotite, and chlorite, were plates with smooth surfaces coated





with clays (Fig. 4). Submicron grains of iron oxides were found at the interfaces between clay coatings and substrates (Fig. 3a, b) or scattered through the clay coatings (Fig. 4c).

### 3.2.3 Intergrade particles

Some dust particles were intergrades between clay-rich particles and clay-coated minerals. The intergrade particles were composed of abundant clays with large mineral inclusions of K-feldspar (Fig. 5a), dolomite (Fig. 5b), and calcite (Fig. 5c), as well as quartz, plagioclase, and large phyllosilicates such as biotite, chlorite, and illite (Supplementary Fig. 1). Clays consisted mostly of ISCMs and kaolinite with submicron grains of iron

(hydr)oxides. The clay matrices of the intergrade particles shown in Fig. 5b and c included palygorskite fibers.

### 3.3 Iron (hydr)oxides

Dust particles dominated by iron (hydr)oxides were rare (Fig. 6). Most iron (hydr)oxides occurred as submicron

grains dispersed in the clay matrices of clay-rich particles (Fig. 7). Although the absolute content of iron (hydr)oxides in mineral dust is small (Lafon et al., 2006; Jeong, 2008; Jeong et al., 2014; Journet et al., 2014), the role of these components in light scattering deserves attention due to their refractive index/absorption (Sokolik and Toon, 1999; Lafon et al., 2006; Kemppinen et al., 2015). The species, volume%, grain size, and distribution of iron (hydr)oxides in the interiors of dust particles therefore must be considered in the modeling of optical

properties.

The mineral species of iron (hydr)oxides identified by electron diffraction and lattice fringes were goethite (Fig. 7a, c, d) and hematite (Fig. 7a, b, e). We could not estimate the relative proportions of goethite and hematite quantitatively because the identification of iron (hydr)oxides was not a routine procedure. Qualitatively, however, goethite grains were likely more common than hematite, similar to the findings of Lafon et al. (2006). Iron

(hydr)oxides were included in 40 of the 48 FIB slices. Goethite was positively identified by electron diffraction and lattice imaging in 19 slices, hematite was identified in eight slices, and magnetite was identified in two slices.

We estimated the total volume of iron (hydr)oxides in the dust particles from their areas in bright-field TEM images, as these minerals occur as dark grains due to their high atomic numbers. However, we could not perform



grain-by-grain identification or measurement of the numerous iron (hydr)oxide grains, as discussed in section 2. The abundance of iron (hydr)oxide grains ranged widely, from high (e.g., 50% in Fig. 7a, 28% Fig. 7e) to moderate (e.g., 15% Fig. 7b, 9% Fig. 7c), sparse (e.g., 3% Figs. 2c and 7d, 0.7% Fig. 3c), and absent (e.g., Figs. 2a and b). The median volume of iron (hydr)oxides in 48 dust particles was 1.5%. Half of the values fell between 1% and 10% (Supplementary Table 1). Some of the dark grains in the TEM images were titanium oxides (Fig. 2b), although their contents were lower than those of iron (hydr)oxides in most particles. Thus, the measured volume% of iron (hydr)oxides must be considered a maximum value, as we could not account for the titanium oxides explicitly.

Grain sizes of iron (hydr)oxides ranged from a few hundred to several tens of nanometers. All longer axes of iron (hydr)oxide grains in the interiors of dust particles fell within the submicron range; most were <0.5 μm with common small grains of several tens of nanometers long (Fig. 7). The grains were well dispersed throughout the clay-rich medium, in contrast to the findings of Deboudt et al. (2012), who detected them more commonly on the surface.

## 3.4 Chemical composition of clay minerals

.

The chemical compositions of clay minerals from the Tenerife dust particles, determined by TEM-EDXS of FIB slices, were plotted on a K–Fe diagram (Fig. 8). The data distribution was consistent with that for Saharan dust collected in Cape Verde (Fig. 6 in Jeong and Achterberg, 2014). Almost all data plotted between ISCMs and kaolinite, confirming the higher kaolinite content of Saharan dust. The average elemental composition of clay minerals, based on 343 EDXS analyses and assuming 14 wt% $H_2O$, was Si 22.4, Al 13.2, Fe 5.0, Mg 1.8, Ti 0.2, K 1.2, Na 0.5, and Ca 0.4 wt%. The average iron content (5.0%) of the clay minerals in the Tenerife dust is consistent with that of the clay minerals in Cape Verde dust. The chemical compositions of ISCMs are plotted in the boxed region of Fig. 8, as derived in Jeong and Achterberg (2014). The average chemical formula of the Saharan dust ISCMs was $K_{0.17}Na_{0.09}Ca_{0.03}(Al_{1.43}Fe^{3+}_{0.38}Mg_{0.36}Ti_{0.01})(Al_{0.50}Si_{3.50})O_{10}(OH)_2$.

## 3.5 Shape of dust particles



The 2D shapes of dust particles have been approximated by ellipses with varying aspect ratios (e.g., Reid et al., 2003). Although the three-dimensional (3D) shapes of the particles resemble ellipsoids, they are – due to the lack of information on the third dimension – usually assumed to be spheroids in particle size analyses (Reid et al., 2003; Kandler et al., 2009; Jeong et al., 2014) and in optical modeling (Mishchenko et al. 1997; Nousiainen and

Vermeulen 2003; Dubovik et al., 2006, Merikallio et al., 2011). As a byproduct of FIB work on studies of dust interiors, the thicknesses of the particles can be derived from TEM images of the cross sections (Fig. 9). The two longer axes ($b$ and $c$) of the ellipsoids were measured from SEM images, and the short axis ($a$) was measured from TEM cross-section images. The average axial ratios of 48 particles, obtained by dividing the lengths of axes $a$ and $c$ by that of axis $b$, were 1.4±0.3 ($c/b$) and 0.5±0.3 ($a/b$), respectively (Supplementary Table 1). The axial

ratios of ellipsoids varied slightly, depending on the major mineral components of the particles. The $a/b$ ratios were highly dependent on the mineralogical types of the particles; the average of this ratio was 0.5 in clay-rich particles; 0.7 in quartz, plagioclase, and K-feldspar with clay coatings; and 0.2 in particles consisting of coarse platy phyllosilicate minerals with clay coatings (Supplementary Table 1). The average of $c/b$ ratio was 1.3 in clay-rich particles, and ~1.5 in coarse minerals with coatings.

### 3.6   Comparison with single-particle properties of Asian dust

The clay content of Saharan dust (81%) is higher than that of Asian dust (57%) (Table 1). Mineral compositions of clays also differ slightly between dusts. The analyzed Saharan dust sample is distinct from Asian dust due to its

higher kaolinite content and lower chlorite content. The average Fe wt% of the clay minerals in Asian dust was 6.7%, which was slightly higher than that for Saharan dust (5%). The lower iron content of Saharan dust clay minerals is consistent with the lower chlorite content and the higher kaolinite content. ISCMs are the major mineral components of both Saharan and Asian dusts. Assuming K fixation in the illite interlayer (Jeong et al., 2004), the proportion of the illitic component of ISCMs was lower in the Tenerife Saharan dust (K = 1.51% on

average) than in Asian dust (K = 2.2%).

The internal structures of the Saharan dust particles are basically similar to those of Asian dust particles, which were grouped into three types: type I, large non-clay grains with clay coatings; type II, clay-rich agglomerates with inclusions of non-clay mineral grains; and type III, large platy minerals with clay coatings.





The dust particles shown in Figs. 2, 5, and 7 have Type II internal structures. Those shown in Figs. 3 and 4 have type I and type III internal structures, respectively. Irregular pores of varying sizes were reported in the interiors of Asian dust particles (Jeong and Nousiainen, 2014). Recent modeling of the optical properties of mineral dust showed that the internal pores of dust particles affect these properties (Kemppinen et al., 2015). Pores in the

interiors of Tenerife Saharan dust particles, however, were rather scarce and small (Figs. 2b, 3c, and 5c; Supplementary Fig. 1). The fine sizes of particles analyzed in this study may explain these findings. No particle observed by SEM exceeded 10 μm (most were < 5 μm), as large particles were rare in the probed dust event. We cannot exclude the possibility that coarse particles have larger pores due to increased heterogeneity from the mixing of different mineral species, grain sizes, and surface properties. TEM analyses of additional dust samples,

particularly those containing coarser particles, are required.

## 3.7   Implications for dust optical properties and micronutrient transport

In most applications, the dust particle optical properties are still based on the assumption of particle homogeneity

(Nousiainen and Kandler, 2015; Kahnert et al., 2016). Based on the analysis of particle interiors of Asian dust by Jeong and Nousiainen (2014), Kemppinen et al. (2015) assessed the impacts of various types of observed inhomogeneity on the dust particles' optical properties. Their conclusion was clear: dust particles exhibit many types of inhomogeneity whose impacts on the optical properties cannot be accurately mimicked by homogeneous particle models. The impact of iron oxides in particular, whether present in isolated grains or mixed within the

clay coating, could not be properly accounted for by any tested means of making the whole particles homogeneous. In addition, the presence of internal cavities led to impacts on optical properties that were challenging to mimic with homogenized particles.

   The problem with iron oxides is that, at solar short-wave frequencies, their complex refractive indices differ substantially from those minerals that make most of the particle volume: the real part of refractive index can be

easily a factor of two larger for the iron oxide, and the imaginary part tend to be larger by many orders of magnitude. A high imaginary part means that the particle interior is strongly absorbing, but the high real part may make it difficult for the radiation to penetrate through the particle surface to be absorbed. However, when a mixing formula is applied to average the refractive indices of the component materials, the typically small





amounts of iron oxides are not sufficient to increase the real part of the refractive index much above that of the dominating minerals; whereas, the extremely large difference in the imaginary parts makes the whole particle interior much more absorptive. As a result, the homogenized particle will absorb light much more efficiently than either an external mixture or a model where the iron oxide grain would be present as an inclusion in a dust

particle. The mixing of constituent minerals into a single, effective refractive index for model dust particles will thus lead to too much absorption and too low single-scattering albedo.

      Saharan dust was found to exhibit many types of inhomogeneity present also in Asian dust. Therefore, it appears plausible that the using homogeneous particle models for Saharan dust would result in similar potential errors as was reported by Kemppinen et al. (2015) for the Asian dust. There are, however, some differences to

note. First, Saharan dust appeared to have fewer and smaller internal pores. Second, their iron oxide content was somewhat smaller. Both of these effects alone suggest a smaller impact on the optical properties due to the inhomogeneity. However, it does not mean that the impacts could be ignored, and it is further possible that the simultaneous decrease in pores and iron oxides changes the impacts.

Therefore, explicit simulations to quantify the impacts are in order, and planned for in the future.

Saharan dust is a carrier of inorganic nutrients to remote ecosystems. Iron is an element limiting phytoplankton growth in remote ocean ecosystems. The modeling of iron supply to global oceans should include mineral compositions and iron contents of constituent minerals, as highlighted in the modeling study of iron biogeochemical cycles conducted by Johnson and Meskhidze (2013). Mineral compositions and their iron chemistries were converted from world soil data for modeling (Claquin et al., 1999; Nikovic et al., 2012; Journet

et al., 2014). However, a large degree of uncertainty remains regarding the mineralogical properties of fine particles in the source soils. In particular, data on the iron content of soil clay minerals are rarely available in the literature. Bulk and single-particle analyses of representative dust provide direct mineralogical information because soil particles lifted from wide source areas have been mixed thoroughly during long-range transport.

      Our TEM analyses also revealed the mineralogical attributes of iron single dust particles. Iron was

partitioned into (hydr)oxides and clay minerals. The iron (hydr)oxides were goethite and hematite, with minor contributions of magnetite. Their median volume contribution to the particles was 1.5 vol%. Iron-bearing silicates are clay mineral aggregates dominated by ISCMs (Table 1). The average iron content of the clay mineral aggregates in Saharan dust particles is 5 wt%. Iron must be dissolved from clay minerals and iron (hydr)oxides



before it is available for phytoplankton growth in the ocean (Rubin et al., 2011; Sholkovitz et al., 2012). The fractional Fe solubility (%Fe$_S$) of mineral dust (Sholkovitz et al., 2012) depends on iron content, crystallinity, and the specific surface areas of iron-bearing minerals (Lasaga, 1995; Nagy, 1995). The common nano-sized grains of ISCMs and iron (hydr)oxides with high specific surface areas and low crystallinity in the dust particles may enhance iron bioavailability (Baker and Jickells, 2006).

## 4. Summary and conclusions

Dust particles are composed of minerals with wide ranges of grain size and optical and chemical properties. The optical properties related to radiative forcing and remote sensing could be calculated optimally based on realistic mineralogical, morphological, and structural models of dust particles. Analysis of micronutrient delivery to remote oceans should consider the forms of the elements in the dust particles. TEM analysis of cross sections of dust particles, aided by EDXS, revealed the details of the internal mineralogical, structural, and morphological properties of fine Saharan dust particles. Saharan dust shares many properties with Asian dust, such as internal structure, mineral composition, and ISCM chemistry, but it differs slightly in its higher clay content, fewer pores, higher kaolinite content, and higher smectitic component in ISCMs. The overall 3D shapes of the dust particles were roughly ellipsoidal, with widely varying axial ratios. We emphasize, however, that this does not necessarily mean that their optical properties would match with those of corresponding ellipsoids. Rather, more likely, they do not (Lindqvist et al., 2014). Submicron grains of iron oxides, mostly goethite and hematite, were commonly dispersed throughout the clay medium. Their median volume% was estimated at ~1.5%. Submicron- to nano-sized grains of ISCMs and iron (hydr)oxides would enhance the release of inorganic nutrients to ecosystems. These data on the mineralogical properties and structures of individual dust particles provide a basis for modeling optical properties of Saharan dust. However, we could not probe the interiors of coarse (> 10 µm) dust particles due to their low abundance in the probed dust plume and the technical difficulty of preparing large cross sections. Description of the general properties of single particles can be completed with coarse particle data from further investigations.





*Acknowledgements*. This study was funded by the National Research Foundation of Korea grant NRF-2011-0028597 to G. Y. Jeong. K. Kandler greatly acknowledges support from the German Research Foundation (DFG, grants KA 2280/2 and FOR 539 SAMUM).

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



Table 1. Mineral number % of Saharan dust based on SEM-EDXS analyses of single particles.

| No. of particle | Total clay minerals | ISCMs[*] | Ka | Ch | Bt | Q | P | Kf | Am | Ca | D | Fe | Ti | Gy | Sp | Ap | Total |
|---|---|---|---|---|---|---|---|---|---|---|---|---|---|---|---|---|---|
| | | | | | | Saharan dust at Tenerife (This study) | | | | | | | | | | | |
| $n$ =1626 | 1319 | 1128 | 176 | 14 | 1 | 164 | 44 | 34 | 2 | 21 | 6 | 11 | 10 | 13 | 1 | 1 | 1626 |
| % | 81.1 | 69.4 | 10.8 | 0.9 | 0.1 | 10.1 | 2.7 | 2.1 | 0.1 | 1.3 | 0.4 | 0.7 | 0.6 | 0.8 | 0.1 | 0.1 | 100 |
| | | | | | Asian dust at Korea (Jeong and Achterberg, 2014)[**] | | | | | | | | | | | | |
| % | 56.1 | 48.4 | 1.8 | 3.9 | 2.1 | 18.7 | 10.5 | 4.0 | 0.7 | 6.5 | 1.0 | 1.2 | 0.2 | 0.8 | 0.2 | 0.2 | 100 |

*ISCM=illite-smectite series clay minerals, Ka=kaolinite, Ch=chlorite, Bt=biotite, Q=quartz, P=plagioclase, Kf=K-feldspar, Am=amphibole, Ca=calcite, D=dolomite, Fe=iron (hydr)oxide, Ti=Ti oxide, Gy=gypsum, Sp=sphene, Ap=apatite.

** Average value of the SEM-EDXS results from three Asian dust samples presented in Jeong and Achterberg (2014). Vaules are slightly differernt from those in Jeong and Achterberg (2014) by the change of mineral species considered for quantification.





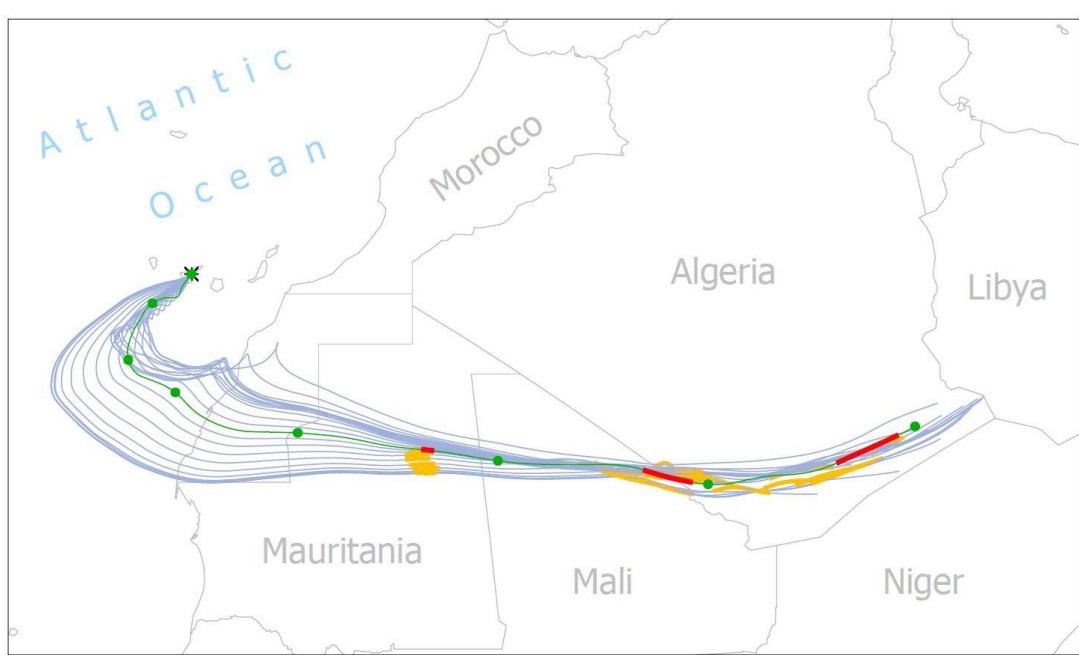

**Figure 1**. Map of northwestern Africa with the hourly 168-h backward trajectory evolution, which shows dust arriving at Izaña on 16 July 2005. Green trajectories represent the impactor sample. The filled circles indicate 24-h periods. Trajectory sections where the altitude is below the mixing layer height are enhanced in orange/red.





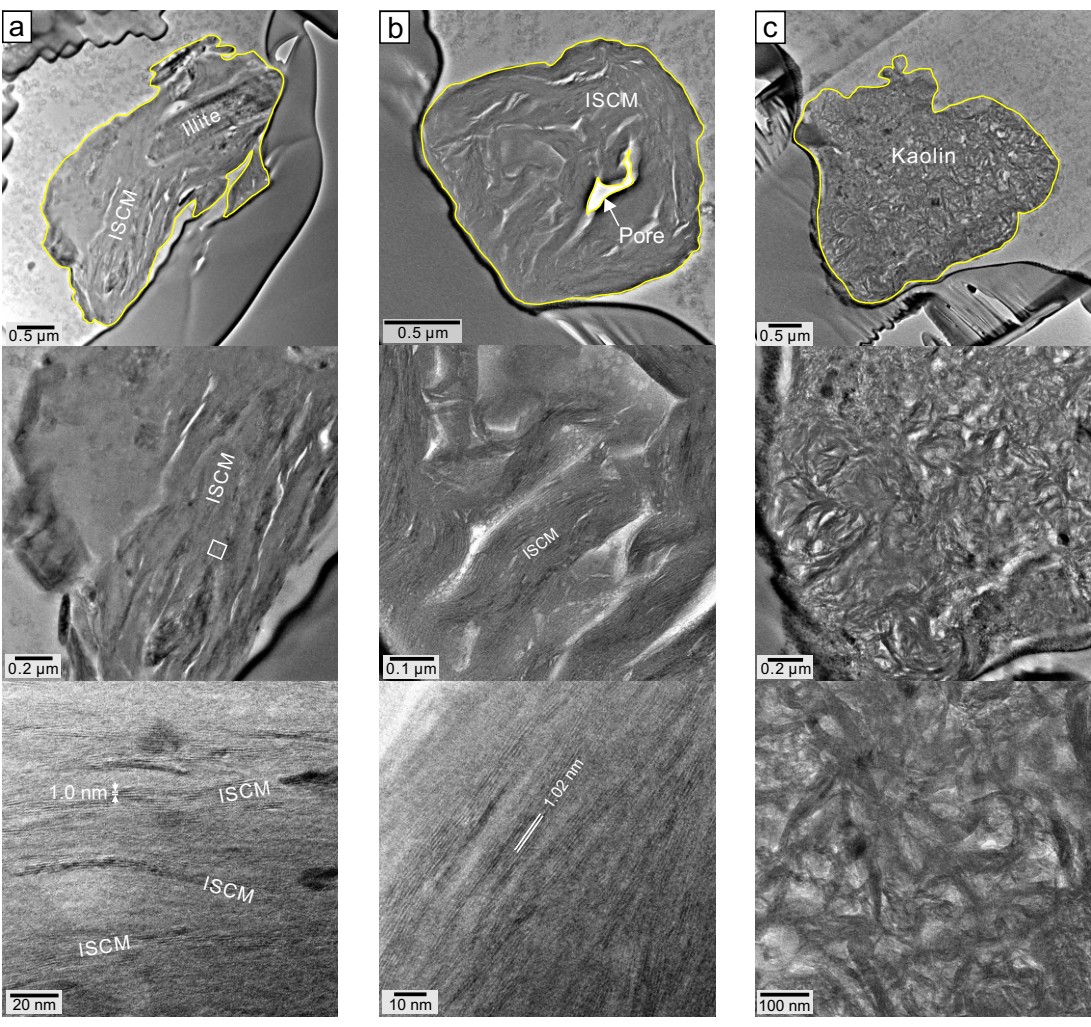

**Figure 2**. TEM images of cross sections of clay-rich dust particles. Yellow lines were added to show particle outlines. ISCM: illite–smectite series clay minerals, including illite, smectite, and their mixed layers. (a) ISCM clays, including an illite grain (particle 30). (b) ISCM clay (particle 48). (c) Kaolin clay (particle 40). Yellow lines were added to show the boundary of particle and pore.



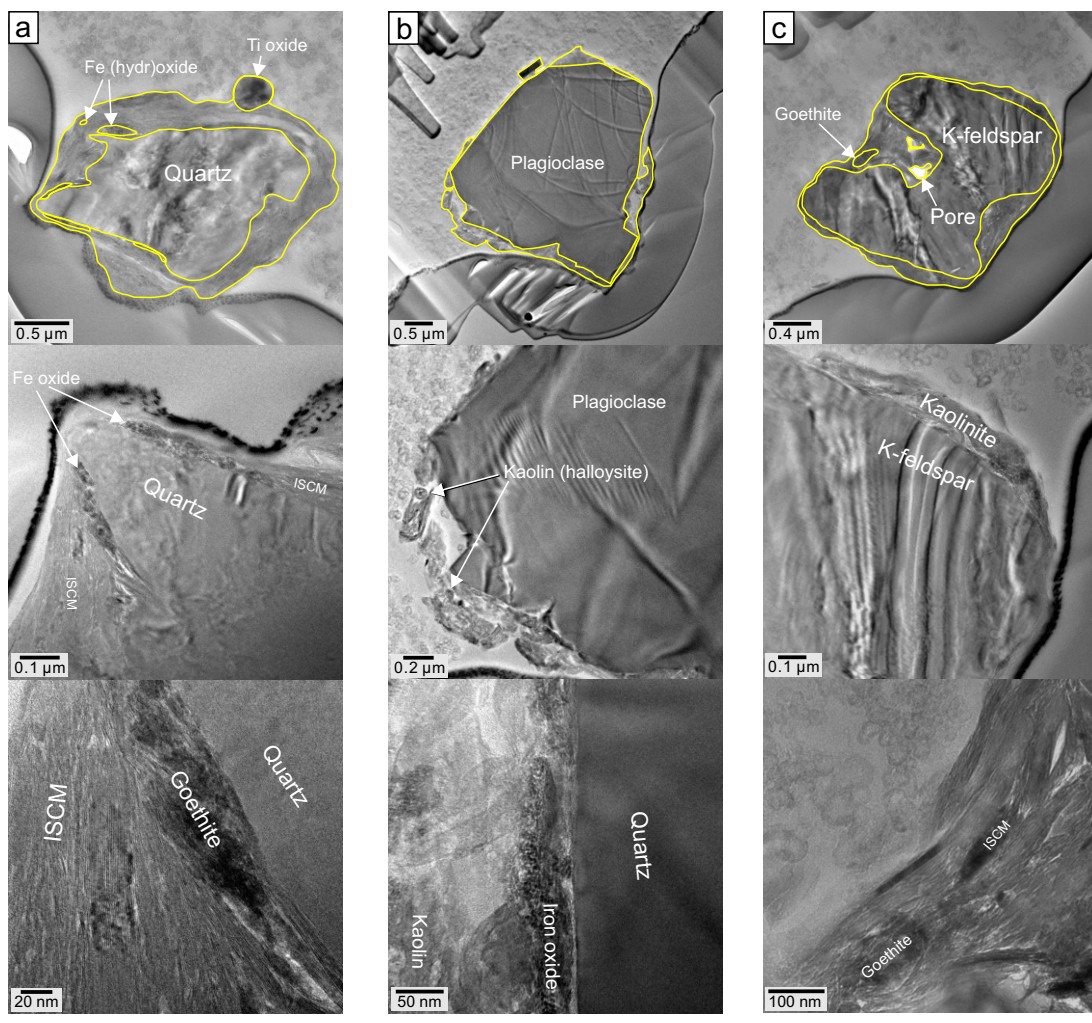

**Figure 3**. TEM images of cross sections of large grains coated with clays. (a) Quartz grain coated with clays (particle 46). (b) Plagioclase grain coated with clays (particle 37). (c) K-feldspar grain coated with clays (particle 28). Yellow lines were added to show particle boundary and the interface between clay coating and substrate.




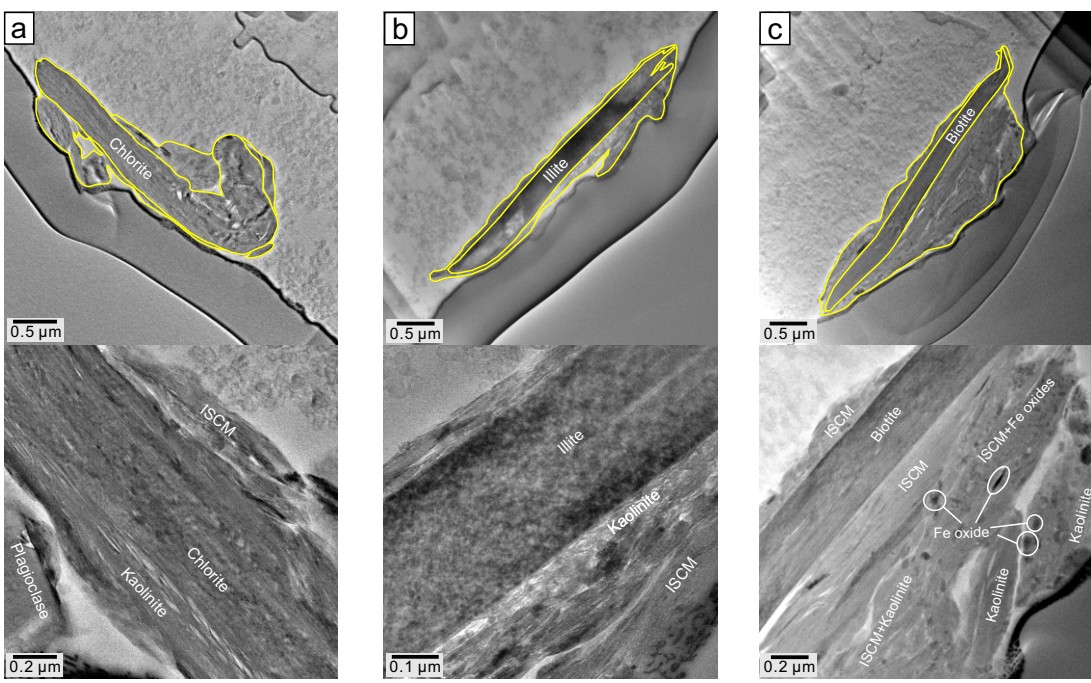

**Figure 4**. TEM images of cross sections of large platy grains coated with clays. (a) Chlorite plate coated with ISCM and kaolinite clays (particle 41). (b) Illite plate coated with ISCM and kaolinite clays (particle 34). (c) Biotite plate coated with ISCM and kaolinite clays (particle 23). Yellow lines were added to show particle boundary and the interface between clay coating and substrate.





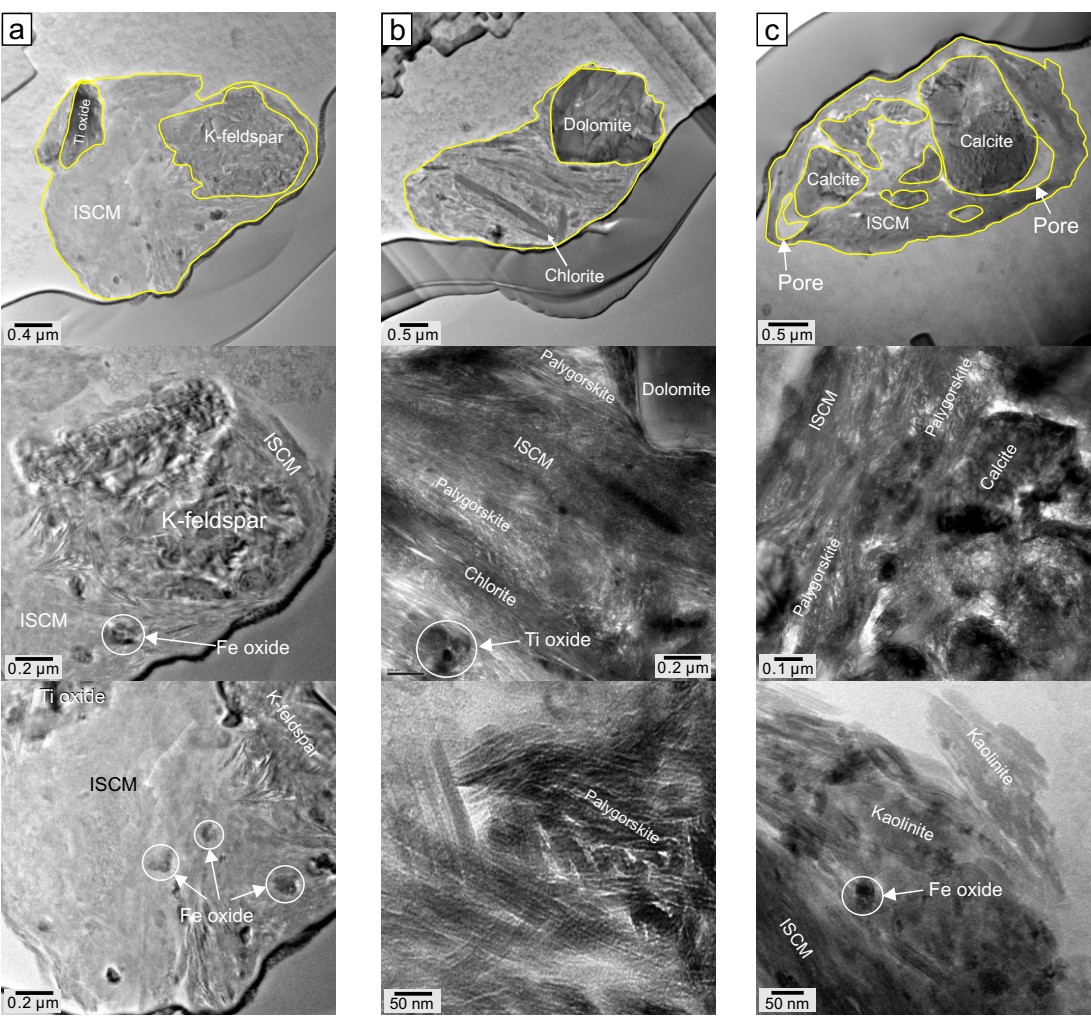

**Figure 5**. TEM images of cross sections of particles composed of clay and non-clay mineral inclusions. (a) K-feldspar and titanium oxide grains agglomerated with ISCM clay (particle 27). (b) Dolomite and chlorite grains agglomerated with ISCM and palygorskite clays (particle 26). (c) Calcite grains agglomerated with ISCM and palygorskite clays (particle 29). Yellow lines were added to show the boundary of particle and pore and the interface between clay and large mineral inclusion.



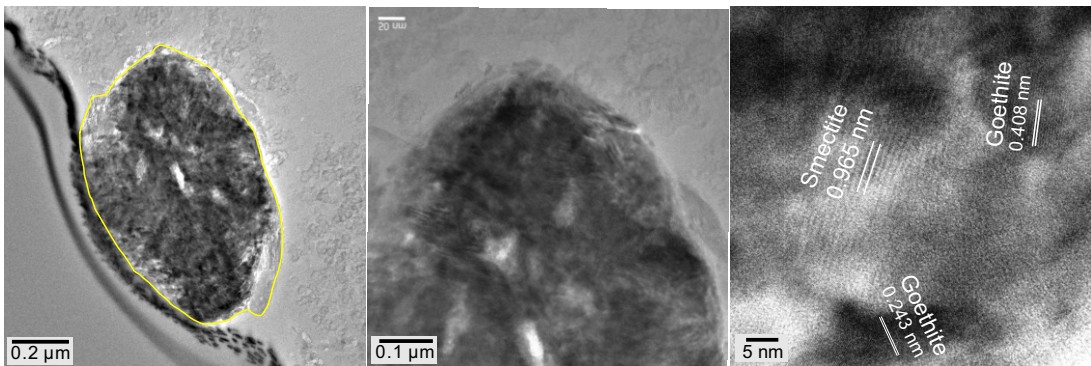

**Figure 6**. TEM images of a cross section of the iron (hydr)oxide (goethite) particle (particle 14). Yellow line was added to show particle boundary.





**Figure 7**. TEM images of cross sections of clay particles containing rich iron (hydr)oxide grains. (a) Mixture of iron (hydr)oxides (goethite and hematite) and kaolinite clay (particle 3). (b) ISCM clay including elongated hematite grains of varying sizes, showing preferred orientation with some titanium oxide grains (particle 43). (c) ISCM-kaolinite clay interspersed with goethite grains of varying sizes (particle 11). (d) ISCM clay including several grains of goethite (particle 21). (e) Platy grains of weathered chlorite showing hematite grains in the cleaved spaces between curved chlorite plates (particle 45). Insets in b and e are X-ray maps.





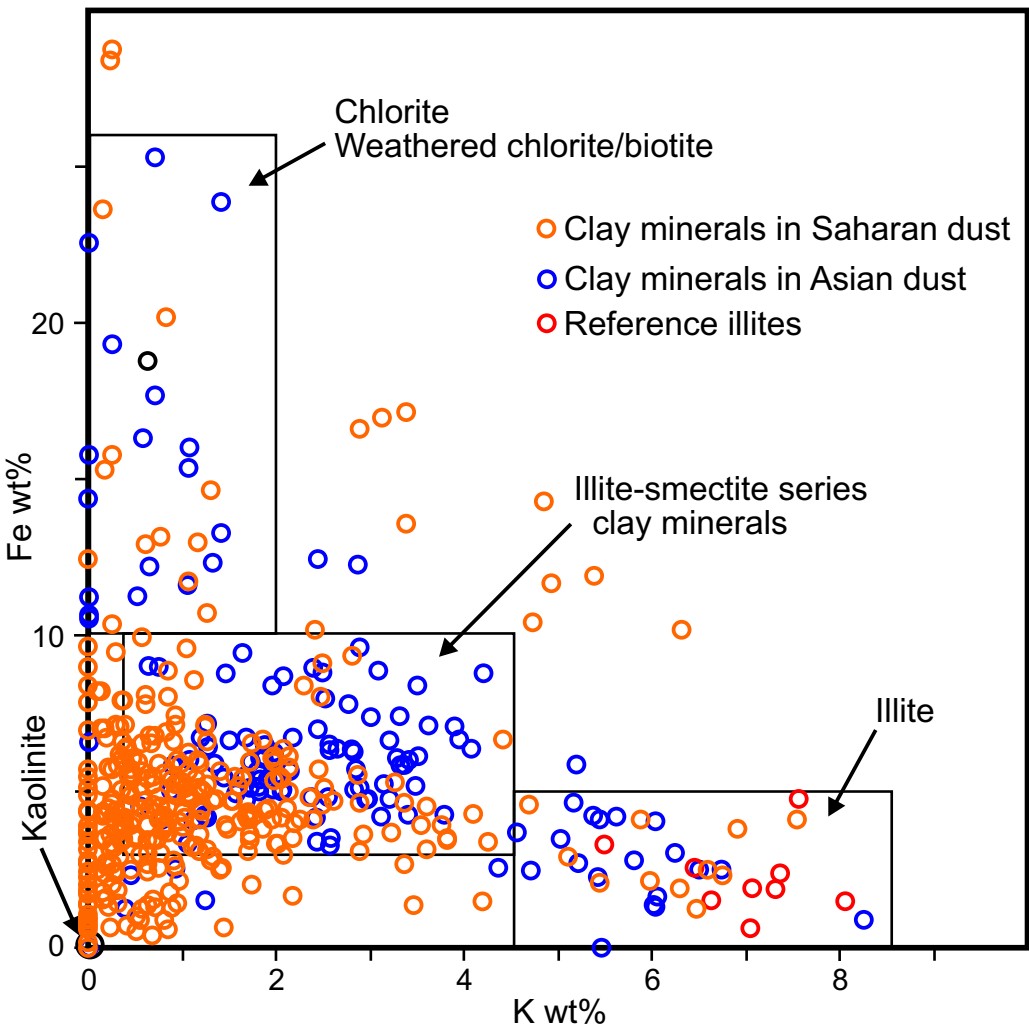

**Figure 8**. Plots of Fe vs. K from the TEM-EDXS analyses of clay minerals in the FIB slices of dust particles. The boxes indicating groups A, B, and C are drawn based on TEM microtextures and EDXS data provided by Jeong and Achterberg (2014).





**Figure 9**. External shapes of 48 dust particles investigated in this study. The left-side sketch of each particle denotes particle outlines observed in the vertical direction by SEM. Two longer axes of dust particle ellipsoids lie parallel to the substrate carbon tape. Right-side sketch denotes particle outline observed in the cross section cut along the straight line of the left-side sketch.