# Peer review of "Mineralogical properties and internal structures of individual fine particles of Saharan dust"

_Atmospheric Chemistry and Physics, 2016_

## Referee Comment (RC1) · L. Shao (Referee) · 19 Jul 2016

This manuscript has investigated the Sahara dust storm particles for their mineralogy and internal textures. The new method, the focused ion beam (FIB) slices, combined with the TEM-EDXS, was used to characterize the internal structures of particles. The iron compounds associated with the dust storm particles were also discussed in terms of their potential relations to the phytoplankton growth in the ocean. Some interesting results have been achieved. The manuscript was well prepared and organized. I recommend this manuscript to be published, with minor revision.

My suggestions are as follows:

1. This manuscript presented a number of mineral species according to the elemental compositions obtained by the TEM-EDXS. However, minerals have general characteristics of 'isomorphism' or 'polymorphism', so we cannot classify the particle mineralogy merely according to the elemental compositions of particles. In this manuscript, the authors have referred to the XRD results and have also used the selected area electronic diffraction as well as lattice fringe imaging. Although these methods of mineral identification was introduced in Jeong and Nousiainen (2014), it is necessary to have a simple description of how to interpret mineral species according to EDXP patterns in this current manuscript. 2. Page 9 section '3.6 Comparison with single-particle properties of Asian dust'ïijŽIt is interesting to see that cay minerals are predominant minerals in the Sahara dust storm particles. This is different from the Asian Dust Storm (ADS) samples and I guess this is mainly due to differences in the particle sizes. Please refer to the paper on the mineralogy of the ADS dust fall and PM10 samples, in which quartz occupied a predominant position (Shao et al., 2007, ADVANCES IN ATMOSPHERIC SCIENCES, VOL. 25, NO. 3, 2008, 395-403). Please check if there are any variations of mineral types in association with the particle sizes between Sahara dust and Asian dust storm particles. Plus, the size- segregated mineral compositions may be more important in characterizing mineralogical properties. 3. Page 6, '3.2 Internal structures of individual dust particles': When discussing the internal structures of individual dust particles, 'Clay-rich particles', 'Large minerals with clay-rich coatings', 'Intergrade particles'. ….., were classified. I am wondering if the term 'aggregate' may be more suitable for these particle categories since the particles presented in this current manuscript are actually the aggregates of minerals, i.e., rock fragments. Not necessary secondary coating. 4. Section "Samples and methods": The methodology needs to be introduced in more detail. A total of 1626 individual particles were analyzed, but how many samples were analyzed? How do you select particles on the filter? Are these samples representative of typical peak dust storm episodes or a non-storm episodes? A table of sample information may be useful. 5. The total clay content determined by XRD for the Cape Verde dust sample was 81%, matching the proportion obtained using single-particle SEM-EDXS data in this study. The value '81%' is a sme value for two cases, too precise!. The XRD measured the volume or weight percentages while

the EDXS give a number percentages, and two methods will not give the same results. Please check this carefully. 6. Page 21, Fig.2c, the 'kaolin' might be 'kaolinite'?

---

## Referee Comment (RC2) · Deboult (Referee) · 29 Aug 2016

**"General comments"**
This paper addresses relevant scientific questions within the scope of ACP. It presents original and interesting works leading to substantial conclusions. It is well structured and clear. In consequent, I recommend its publication with minor revision.

**"Specific comments"**
1- Several samples were collected, but considering the technical difficulties for sample preparation, one sample was only studied for its mineralogical properties and internal structures (collected the 15 July 2005 at 9:33). How was it chosen among all samples? Which impaction stages were selected? Why? What was the PM2.5 concentration during sampling? It should be more precisely exposed in the "sample and methods" part.

2- Criteria used for the identification of predominant minerals both from EDX data and electron diffraction/lattice fringe imaging should be presented in supporting information.

3- In the section 3.1, the relative abundance of clay mineral particles obtained from SEM-EDXS data (Table 1) for the Tenerife sample is directly compared with the total clay content (in wt%?) determined by XRD for a Cape Verde sample, but they are two different physical parameters. This point should be clarified.

4- In the section 3.3, the volume of iron (hydr)oxides in the dust particles is estimated. Two dimensions of the iron grains in dust are obtained from the bright-field TEM images, but how is measured the third dimension, notably for internal grains? From EDX maps? It should be explained.

5- The section "3.5 Shape of dust particles" notably presents the methodology to measure the total particle volume, so it should be inserted before the part 3.3 in which particle volumes are presented and also discussed (sup Table 1).

**"Technical corrections"**
6- (Avila et al., 1997) is cited in the introduction, but (Avila et al., 1996) is listed in the bibliography.

7- (Conny, 2013) is cited in the introduction, but it is not among the listed references.

8- Figure 1: What is the level high of the arrival air masses for the backward trajectory calculations by HYSPLIT? It should be mentioned in the Figure caption.

9- Figure 2: The meaning of yellow lines is given two times in the Figure caption.

---

## Author Comment (AC1) · 13 Sep 2016

**Reply to the comments by L. Shao**

We appreciate the valuable comments.

**Comment (1)** This manuscript presented a number of mineral species according to the elemental compositions obtained by the TEM-EDXS. However, minerals have general characteristics of 'isomorphism' or 'polymorphism', so we cannot classify the particle mineralogy merely according to the elemental compositions of particles. In this manuscript, the authors have referred to the XRD results and have also used the selected area electronic diffraction as well as lattice fringe imaging. Although these methods of mineral identification was introduced in Jeong and Nousiainen (2014), it is necessary to have a simple description of how to interpret mineral species according to EDXP patterns in this current manuscript.

**Reply (1)** The procedures of mineral identification using EDXS, lattice fringe imaging, and electron diffraction are now provided in the supplement information. We agree that mineral identification should be conservative because of isomorphism or polymorphism. It is impossible to identify all the minerals on the species level. Identification of calcite, dolomite, and quartz are straightforward on the basis of TEM-EDXS spectra. However, we cannot separately identify three $KAlSi_3O_8$ polymorph mineral species including sanidine, orthoclase, and microcline soley on the basis of TEM-EDXS pattern. Even the lattice fringe imaging and electron diffraction pattern are not straightforward to polymorph identification because of the similar crystal structures. Since the purpose of our study is not in-depth crystal-structural study of each mineral, we group three $KAlSi_3O_8$ polymorphs into K-feldspar. Chlorite and plagioclase are also mineral groups but not species. Without the aid of electron diffraction and lattice fringe imaging, we can identify plagioclase based on TEM-EDXS spectra easily, but the identification of plagioclase mineral species including albite, oligoclase, andesine,... is not straightforward because we have to know the accurate ratio of Ca and Na. Frequent Na loss during the TEM operation and FIB sample preparation inhibits the precise determination of plagioclase chemical composition. Identification of nano-thin illite, smectite, illite-smectite mixed layers is also challenging as discussed in the text and Jeong and Nousiainen (2014), resulting in grouping them into illite-smectite series clay minerals (ISCMs). Although TEM is a powerful tool for the identification of fine mineral grains in the dust particles in comparison to SEM and XRD, identification is limited in many cases because of beam damage and high vacuum.

**Changes in the manuscript (1)** The procedures of mineral identification using EDXS, lattice fringe imaging, and electron diffraction are **now provided in the supplement information as attached at the end of this letter**.

**Comment (2)** Page 9 section '3.6 Comparison with single-particle properties of Asian dust is interesting to see that cay minerals are predominant minerals in the Sahara dust storm particles. This is different from the Asian Dust Storm (ADS) samples and I guess this is mainly due to differences in the particle sizes. Please refer to the paper on the mineralogy of the ADS dust fall and PM10 samples, in which quartz occupied a predominant position (Shao et al., 2007, ADVANCES IN ATMOSPHERIC SCIENCES, VOL. 25, NO. 3, 2008, 395-403). Please check if there are any variations of mineral types in association with the particle sizes between Sahara dust and Asian dust storm particles. Plus, the size-segregated mineral compositions may be more important in characterizing mineralogical properties.

**Reply (2)** We agree that mineralogical properties are related to the particle size. We hope to analyze coarse particles of Saharan dust (> 10 μm) by XRD and SEM in future if samples are available. Our study was focused on the clarification of the internal structures and mineralogy of Saharan dust

particles. In the original manuscript, we presented the mineral composition of total dust particles from two impact stages. Table 1 was revised by adding the mineral compositions of two impact stages (0.9–2.6 µm and > 2.6 µm) as shown below. Dust particles of the two size fractions were dominated by clay minerals including ISCMs and kaolinite. Dust particles of 0.9–2.6 µm in size, however, were slightly enriched with clay minerals in comparison with those of > 2.6 µm in size, while coarser particles (> 2.6 µm) were rather enriched with non-clay minerals including quartz, plagioclase, and K-feldspar.

**Changes in the manuscript (2)** We changed the first paragraph of the section 2 Samples and methods to "Dust samples used in this study collected from the top of the tower building of the meteorological observatory of Izaña, Tenerife, Spain (28° 18' 33.8" N, 16° 29' 56.9" W, 2395 m a.s.l.). Details on the procedure and location can be found in Kandler et al. (2007). Particles were collected with a cascade impactor on carbon adhesive, with nominal stage size ranges of > 2.6 µm, 0.9–2.6 µm, and 0.1–0.9 µm (50 % efficiency cut-off aerodynamical particle diameter), of which the stages with > 0.9 µm were used for analysis. All samples were stored in a desiccator under dry conditions prior to analysis. The sample analyzed in the present work was collected during a 10 min period on 15 July 2005 at 09:33 h (UTC). It was selected as best-representing the campaign to achieve the greatest atmospheric relevance. Its composition is close to the mean campaign composition (cf. Kandler et al., 2007, Fig. 8), and the corresponding transport trajectory is central in the observed trajectory field (Fig. 1 and Kandler et al., 2007, Fig. 9). According to AERONET aerosol optical thickness data (Supplement Fig. 1), the concentration on the sampling day was also close to the average of the dust event lasting from July 12 to July 22, 2005. Analysis was limited to particles with diameter < 5 µm, as too few larger ones were available"

SEM-EDXS results of two impact stages were added to Table 1.

Table 1. Mineral number % of Saharan dust based on SEM-EDXS analyses of single particles.

| Size | No. of particle | Total clay minerals | ISCMs[*] | Ka | Ch | Bt | Q | P | Kf | Am | Ca | D | Fe | Ti | Gy | Sp | Ap |
|---|---|---|---|---|---|---|---|---|---|---|---|---|---|---|---|---|---|
| Saharan dust, Tenerife (This study) | | | | | | | | | | | | | | | | | |
| 0.9-2.5 µm | n=1191 | 1005 | 868 | 131 | 6 | 0 | 105 | 21 | 17 | 1 | 12 | 5 | 8 | 8 | 9 | 0 | 0 |
| | % | 84.4 | 72.9 | 11.0 | 0.5 | 0.0 | 8.8 | 1.8 | 1.4 | 0.1 | 1.0 | 0.4 | 0.7 | 0.7 | 0.8 | 0.0 | 0.0 |
| > 2.5 µm | n=435 | 314 | 260 | 45 | 8 | 1 | 59 | 23 | 17 | 1 | 9 | 1 | 3 | 2 | 4 | 1 | 1 |
| | % | 72.7 | 60.2 | 10.4 | 1.9 | 0.2 | 13.7 | 5.3 | 3.9 | 0.2 | 2.1 | 0.2 | 0.7 | 0.5 | 0.9 | 0.2 | 0.2 |
| Total | n=1626 | 1319 | 1128 | 176 | 14 | 1 | 164 | 44 | 34 | 2 | 21 | 6 | 11 | 10 | 13 | 1 | 1 |
| | % | 81.1 | 69.4 | 10.8 | 0.9 | 0.1 | 10.1 | 2.7 | 2.1 | 0.1 | 1.3 | 0.4 | 0.7 | 0.6 | 0.8 | 0.1 | 0.1 |
| Asian dust, Korea (Jeong and Achterberg, 2014)[**] | | | | | | | | | | | | | | | | | |
| | % | 56.1 | 48.4 | 1.8 | 3.9 | 2.1 | 18.7 | 10.5 | 4.0 | 0.7 | 6.5 | 1.0 | 1.2 | 0.2 | 0.8 | 0.2 | 0.2 |

*ISCM=illite-smectite series clay minerals, Ka=kaolinite, Ch=chlorite, Bt=biotite, Q=quartz, P=plagioclase, Kf=K-feldspar, Am=amphibole, Ca=calcite, D=dolomite, Fe=iron (hydr)oxide, Ti=Ti oxide, Gy=gypsum, Sp=sphene, Ap=apatite.
** Average value of the SEM-EDXS results from three Asian dust samples presented in Jeong and Achterberg (2014). Vaules are slightly differernt from those in Jeong and Achterberg (2014) by the change of mineral species considered for quantification.

We added and revised Section 3.1 Mineral composition of bulk dust: "Single-particle SEM-EDXS data showed that dust particles in both size fractions were dominated by clay minerals including ISCMs and kaolinite. Dust particles of 0.9–2.6 µm in size, however, were slightly enriched in clay minerals in comparison with those of > 2.6 µm in size, while non-clay minerals including quartz, plagioclase, and K-feldspar were rather enriched in coarser particles (> 2.6 µm ). Average mineral composition was ISCMs 69%, kaolinite 11%, quartz 10%, plagioclase 3%, K-feldspar 2%, calcite 1%, chlorite 1%, iron (hydr)oxide 1%, titanium oxide 1%, and gypsum 1% (Table 1). The total clay

mineral content was 81%."

We added sentences to the section 3.6 Comparison with single-particle properties of Asian dust: "Previous data showed that total clay content of Saharan dust ranged from 61% to 73% in the samples of the Atlantic islands (Glaccum and Prospero,1980), from 56% to 81% in the dustfall collected in Spain (Avila et al., 1997), and around 81% in two Saharan dust samples collected in Cape Verde (Jeong and Achterberg, 2014). Total clay content of Asian dust ranged from 28% to 50% in Beijing, China (Shao et al., 2008) and around 57% in Korea (Jeong and Achterberg, 2014). Thus, clay minerals are likely enriched in Saharan dust in comparison with Asian dust."

Shao et al. (2007) was added to the list of reference.

**Comment (3)** Page 6, '3.2 Internal structures of individual dust particles': When discussing the internal structures of individual dust particles, 'Clay-rich particles', 'Large minerals with clay-rich coatings', 'Intergrade particles': : :.., were classified. I am wondering if the term 'aggregate' may be more suitable for these particle categories since the particles presented in this current manuscript are actually the aggregates of minerals, i.e., rock fragments. Not necessary secondary coating.

**Reply (3)** Our manuscript was written assuming readers majorly working in the fields of atmospheric chemistry, environment, and light scattering where mineral dusts are treated as airborne 'particles' of ranging sizes and mineralogy. Of course, the particles are agglomerates (or aggregates) of smaller mineral grains. In the text an individual airborne solid object was termed "particle" and a particle constituent was termed "grain" (so particles are agglomerates of small mineral grains).

**Changes in the manuscript (3)** We would like to keep the term 'particle'.

**Comment (4)** Section "Samples and methods": The methodology needs to be introduced in more detail. A total of 1626 individual particles were analyzed, but how many samples were analyzed? How do you select particles on the filter? Are these samples representative of typical peak dust storm episodes or a non-storm episodes? A table of sample information may be useful.

**Reply (4)** One dust sample during a dusty period lasting from July 12[th] to July 22[nd], 2005 (Kandler et al., 2007) was analyzed in this study. There was no local aerosol concentration measurement, but according to data available from AERONET and WDCA, the day of sampling (July 15[th], 2005) had an average AOT. The sample was selected to be closest to the campaign composition average and the central transport trajectory. This is described in the modified manuscript now. So, the sample can be regarded as representative for a medium intensity dust event.

[Figure]

[Figure]

0.9 – 2.6 μm size fraction                              > 2.6 μm size fraction

For SEM-EDXS analyses, we selected rectangular areas where dust particles were homogeneously distributed as shown in two SEM images above. Then, we analyzed all the particles within the area. We have analyzed 1191 and 435 dust particles from impact stage 0.9–2.6 µm and > 2.6 µm, respectively. Total number of analyzed particles was 1626. We have presented the SEM-EDXS result of total 1626 dust particles in the Table 1 of original manuscript. We added SEM-EDXS results of two impact stages to the revised Table 1 as shown above. The data show that fine dust particles were more enriched in clays.

Since FIB slicing cannot be applied to all the particles, 21 and 27 dust particles were selected for FIB work from the size ranges of 0.9–2.6 µm and > 2.6 µm, respectively. Dust particles for FIB work was selected to reflect the mineral abundance of the bulk dust determined by SEM-EDXS single particle analyses. Thus, we think that the internal structures found in this study are representative of the dust particles. The experimental section was revised to clarify particle selection.

**Changes in the manuscript (4)** We added to the revised manuscript a plot on aerosol optical thickness of July 2005 to Supplement Fig. 1.

[Figure]

Supplement Fig. 1: Aerosol Robotic Network aerosol optical thickness for Izana, July 2005. Begin and end of the July dust period is marked by orange bars, the sampling day is indicated by a red arrow. Figure created by http://aeronet.gsfc.nasa.gov/cgi-bin/bamgomas_interactive on September 6th, 2016.

We added two paragraphs to the section 2 Samples and Methods:

"All the particles collected on the impact stages were not analyzed. We selected rectangular areas of homogeneous particle distribution at low magnification, and then, EDXS patterns were obtained from all the particles in the areas. Total number of the analyzed dust particles was 1626: 1191 particles from the impact stage 0.9–2.6 µm, and 435 particles from the stage > 2.6 µm)."

"The mineral composition obtained by this method is evidently semi-quantitative, considering large inherent uncertainty. Previous analyses of Asian dust, however, showed that SEM-EDXS results were consistent with XRD results (Park and Jeong, 2016). The 48 target particles for FIB work were selected according to the mineralogical types of the dust particles, reflecting mineral abundance of bulk dust determined by EDXS analyses: 21 particles from the impact stage 0.9–2.6 µm and 27 particles from the impact stage > 2.6 µm, respectively."

**Comment (5)** The total clay content determined by XRD for the Cape Verde dust sample was 81%, matching the proportion obtained using single-particle SEM-EDXS data in this study. The value '81%'

is a sme value for two cases, too precise!. The XRD measured the volume or weight percentages while the EDXS give a number percentages, and two methods will not give the same results. Please check this carefully.

**Reply (5)** Dust samples for TEM and SEM analyses in this study were collected in the Tenerife, Canary Islands. Thus, complete matching between XRD data of Cape Verde dust and SEM data Tenerife dust is impossible, and only accidental. Close matching of analytical values of two dust samples support the reliability of our XRD and SEM-EDXS quantification procedure applied to minute dust samples. Nevertheless both data are incomplete because of the lack of either XRD or SEM-EDXS data. This has been caused by unavailability of samples suitable for XRD in Tenerife and SEM-EDXS single particle analysis in Cape Verde (sampling campaign were done several years ago).

**Changes in the manuscript (5)** Thus, we deleted the comparison of Tenerife SEM-EDXS data to Cape Verde XRD data.

**Comment (6)** Page 21, Fig.2c, the 'kaolin' might be 'kaolinite'?

**Reply and Changes in the manuscript (6)** The 'kaolin' in Fig.2c was replaced by 'kaolinite' in the revised version.

Supplement Information will be included in the revised manuscript.

**Supplement Information – Mineral identification**

Minerals have their own crystal structures and chemical compositions. Thus, mineral identification using TEM is based on the lattice-fringe imaging and electron diffraction providing structural information and EDXS providing chemical information. Precise identification of all the minerals in the FIB slice to the species level is practically impossible because of beam damage, high vacuum, lower reliability of lattice fringes/electron diffraction data in comparison with XRD, and enormous time required.

**TEM identification of nonphyllosilicate minerals**

The identification of quartz, K-feldspar, plagioclase, calcite, amphibole, dolomite, titanite, apatite, and gypsum was straightforward based on their characteristic EDX spectra (Fig. S1). Although we could not identify mineral species of K-feldspar (sanidine, orthoclase, microcline), plagioclase (albite, oligoclase, andesine...), and amphibole (tremolite, actinolite, hornblende...) using time-consuming complex operation, the purpose of current mineral dust research was satisfied by grouping similar mineral species. Silica phase of the mineral dust from desert was almost quartz, consistent with XRD although few amorphous silica was identified by electron diffraction.

[Figure]

Fig. S1. TEM EDXS patterns of constituent minerals of dust particles obtained from the cross-sectional slices prepared by FIB.

**TEM identification of phyllosilicate minerals**

Phyllosilicate minerals were abundant in the mineral dust. The identification of muscovite, biotite, and chlorite was rather straightforward from their characteristic chemical compositions with the aid of lattice-fringe imaging (Fig. S2). However, the identification of nano-thin phyllosilicates (clay minerals) was difficult because of their breakdown under electron beam and small grain size below the minimum diameter of electron beam for EDXS. They occurred often as agglomerates. In addition, mixed layering of illite and smectite is common in natural soils. The identification of clay minerals was based on lattice fringes and chemical compositions: 1.0 nm for illite, ~1.0 nm for smectite and vermiculite, and ~7.0 nm for kaolinite (Fig. S2). Kaolinite was directly identified from its EDXS with the aid of lattice fringe imaging. However, illite, smectite, and illite-smectite mixed layers could not be separately identified each other because smectite was contracted under the high vacuum of the TEM chamber, showing ~1.0 nm lattice fringes similar to those of illite. Although EDXS can be used for identifying illite and smectite with interlayer cations K and Ca, respectively, they cannot be separately analyzed using EDXS, even when using an electron beam as small as possible. Therefore, we could not distinguish between nano-thin illite, smectite, and their mixed-layers, using conventional TEM work. To avoid over-interpretation, nano-thin platelets of clay minerals showing ~1.0 nm lattice fringes with varying K and Ca contents were grouped into illite-smectite series clay minerals (ISCMs). ISCMs are nano-scale mixtures of nano-thin platelets of illite, smectite, and illite-smectite mixed-layers.

[Figure]

Fig. S2. Identification of phyllosilicates using TEM-EDXS and lattice fringes.

**TEM identification of iron (hydr)oxides**

    Mineralogical identification of iron (hydr)oxides was also challenging. EDXS could not be used for the identification. Electron diffraction and lattice-fringe imaging should be used in combination as shown in Fig. S3. However, many iron (hydr)oxide grains could not be identified because of the overlap of many *d*-spacings, varying crystallographic orientation, and tiny grain sizes. Thus, we used species names only in cases in which mineral species were identified unambiguously by lattice fringe imaging and electron diffraction; in other cases, we used the collective term "iron (hydr)oxide".

[Figure]

Fig. S3. Identification of phyllosilicates using lattice fringes and electron diffraction.

**Mineralogical classification of dust particles using SEM-EDXS**

    Dust particles are essentially mixtures of mineral grains of diverse species and sizes. In case the quantity of powder dust samples is sufficient (~several hundred mg), XRD method is best for the determination of mineral composition. SEM-EDXS analyses of individual particle can be used when powder samples are insufficient or non-available. Ideally, mineral composition of individual dust particle can be determined by mixing several minerals to get the overall chemical composition of the particle. Then, the summation of the mineral compositions of thousands of dust particles considering their volume would lead to the mineral composition of bulk dust. However, the irregular morphology of dust particles prohibits the accurate determination of dust particles due to the difficulty of calibration. In addition, the chemical compositions of constituent minerals are varied. Prior to the development of reliable quantitative analysis procedure based on SEM for the mineral composition of individual dust particle, we adopted semi-quantitative approach. Since dust particles are generally dominated by one mineral species or group, we have determined the predominant mineral of a dust particle referring to the EDXS patterns of pure minerals as shown in Figs. S1 and S2. In case particles show intermediate EDXS pattern (Fig. S4), half of the particle was counted (0.5). Summation of the counts led to the approximate mineral composition of bulk dust. Although the procedure is evidently semi-quantitative, SEM-EDXS results were consistent with XRD results in the recent analyses of Asian dust (Table 1 in Park and Jeong (2016), Journal of the Mineralogical Society of Korea, 29, 79–87).

**SEM-EDXS**

[Figure]

Fig. S4. SEM-EDXS of dust particles.

Park and Jeong (2016)

**Table 1.** Mineral compositions of Asian dusts determined by XRD analysis and SEM-EDS single particle analysis

| Minerals | Asian dust XRD | | | | |
| --- | --- | --- | --- | --- | --- |
| | Feb 22 2015 | Mar 18 2014 | Mar 31 2012 | Mar 20 2010 | Average |
| | (This study) | (Jeong and Achterberg, 2014) | | | |
| ISCMs | 55 | 60 | 42 | 50 | 51 |
| Kaolinite | 2 | 1 | 3 | 4 | 3 |
| Chlorite | 5 | 3 | 6 | 7 | 5 |
| *Total clay* | *62* | *64* | *52* | *61* | 59 |
| Quartz | 18 | 14 | 23 | 15 | 17 |
| Plagioclase | 10 | 11 | 15 | 10 | 12 |
| K-feldspar | 4 | 0 | 6 | 2 | 3 |
| Amphibole | 0 | 0 | 1 | 2 | 1 |
| Calcite | 5 | 5 | 2 | 5 | 4 |
| Gypsum | 1 | 6 | 2 | 6 | 4 |
| Total | 100 | 100 | 100 | 100 | 100 |
| | SEM single particle analysis | | | | |
| ISCMs | 57 | 54 | 48 | 54 | 52 |
| Kaolinite | 2 | 1 | 3 | 2 | 2 |
| Chlorite | 3 | 2 | 4 | 6 | 4 |
| *Total clay* | *62* | *58* | *55* | *62* | 58 |
| Quartz | 19 | 19 | 21 | 17 | 19 |
| Plagioclase | 9 | 11 | 11 | 10 | 11 |
| K-feldspar | 3 | 4 | 5 | 3 | 4 |
| Amphibole | 0 | 1 | 1 | 0 | 1 |
| Calcite | 4 | 7 | 7 | 6 | 7 |
| Gypsum | 2 | 0 | 1 | 1 | 1 |
| Total | 100 | 100 | 100 | 100 | 100 |

We will prepare final version considering comment and reply above.

Sincerely

On behalf of co-authors

Gi Young Jeong
Corresponding Author

---

## Author Comment (AC2) · 13 Sep 2016

**Reply to the comments by Karine Deboudt**

We appreciate the valuable comments.

**Comment (1)** Several samples were collected, but considering the technical difficulties for sample preparation, one sample was only studied for its mineralogical properties and internal structures (collected the 15 July 2005 at 9:33). How was it chosen among all samples? Which impaction stages were selected? Why? What was the PM2.5 concentration during sampling? It should be more precisely exposed in the "sample and methods" part.

**Reply (1)** The sample was chosen due to its closeness to the campaign average in terms of composition and trajectories. Particles were analyzed from the super-micron impaction stages, as they are most complex with respect to optical properties. We have added the corresponding description to the manuscript. Mass concentrations are unfortunately not available for the campaign, but AOT data from aerosol and WDCA shows that the sample was taken during a medium AOT.

**Changes in manuscript (1)** Corresponding information is provided in the supplement Fig. 1.

[Figure]

Supplement Fig. 1: Aerosol Robotic Network aerosol optical thickness for Izana, July 2005. Begin and end of the July dust period is marked by orange bars, the sampling day is indicated by a red arrow. Figure created by http://aeronet.gsfc.nasa.gov/cgi-bin/bamgomas_interactive on September 6th, 2016.

We changed the first paragraph of the section 2 Samples and methods to **"**Dust samples used in this study collected from the top of the tower building of the meteorological observatory of Izaña, Tenerife, Spain (28° 18' 33.8" N, 16° 29' 56.9" W, 2395 m a.s.l.). Details on the procedure and location can be found in Kandler et al. (2007). Particles were collected with a cascade impactor on carbon adhesive, with nominal stage size ranges of > 2.6 µm, 0.9–2.6 µm, and 0.1–0.9 µm (50 % efficiency cut-off aerodynamical particle diameter), of which the stages with > 0.9 µm were used for analysis. All samples were stored in a desiccator under dry conditions prior to analysis. The sample analyzed in the present work was collected during a 10 min period on 15 July 2005 at 09:33 h (UTC). It was selected as best-representing the campaign to achieve the greatest atmospheric relevance. Its composition is close to the mean campaign composition (cf. Kandler et al., 2007, Fig. 8), and the corresponding transport trajectory is central in the observed trajectory field (Fig. 1 and Kandler et al., 2007, Fig. 9). According to AERONET aerosol optical thickness data (Supplement Fig. 1), the concentration on the sampling day was also close to the average of the dust event lasting from July 12 to July 22, 2005.

Analysis was limited to particles with diameter < 5 μm, as too few larger ones were available**"**

SEM-EDXS results of two impact stages were added to Table 1.

Table 1. Mineral number % of Saharan dust based on SEM-EDXS analyses of single particles.

| Size | No. of particle | Total clay minerals | ISCMs* | Ka | Ch | Bt | Q | P | Kf | Am | Ca | D | Fe | Ti | Gy | Sp | Ap |
|---|---|---|---|---|---|---|---|---|---|---|---|---|---|---|---|---|---|
| colspan | | | | | | | Saharan dust, Tenerife (This study) | | | | | | | | | | |
| 0.9-2.5 μm | n=1191 | 1005 | 868 | 131 | 6 | 0 | 105 | 21 | 17 | 1 | 12 | 5 | 8 | 8 | 9 | 0 | 0 |
| | % | 84.4 | 72.9 | 11.0 | 0.5 | 0.0 | 8.8 | 1.8 | 1.4 | 0.1 | 1.0 | 0.4 | 0.7 | 0.7 | 0.8 | 0.0 | 0.0 |
| > 2.5 μm | n=435 | 314 | 260 | 45 | 8 | 1 | 59 | 23 | 17 | 1 | 9 | 1 | 3 | 2 | 4 | 1 | 1 |
| | % | 72.7 | 60.2 | 10.4 | 1.9 | 0.2 | 13.7 | 5.3 | 3.9 | 0.2 | 2.1 | 0.2 | 0.7 | 0.5 | 0.9 | 0.2 | 0.2 |
| Total | n=1626 | 1319 | 1128 | 176 | 14 | 1 | 164 | 44 | 34 | 2 | 21 | 6 | 11 | 10 | 13 | 1 | 1 |
| | % | 81.1 | 69.4 | 10.8 | 0.9 | 0.1 | 10.1 | 2.7 | 2.1 | 0.1 | 1.3 | 0.4 | 0.7 | 0.6 | 0.8 | 0.1 | 0.1 |
| colspan | | | | | | Asian dust, Korea (Jeong and Achterberg, 2014)** | | | | | | | | | | | | |
| % | | 56.1 | 48.4 | 1.8 | 3.9 | 2.1 | 18.7 | 10.5 | 4.0 | 0.7 | 6.5 | 1.0 | 1.2 | 0.2 | 0.8 | 0.2 | 0.2 |

*ISCM=illite-smectite series clay minerals, Ka=kaolinite, Ch=chlorite, Bt=biotite, Q=quartz, P=plagioclase, Kf=K-feldspar, Am=amphibole, Ca=calcite, D=dolomite, Fe=iron (hydr)oxide, Ti=Ti oxide, Gy=gypsum, Sp=sphene, Ap=apatite.
** Average value of the SEM-EDXS results from three Asian dust samples presented in Jeong and Achterberg (2014). Vaules are slightly differernt from those in Jeong and Achterberg (2014) by the change of mineral species considered for quantification.

**Comment (2)** Criteria used for the identification of predominant minerals both from EDX data and electron diffraction/lattice fringe imaging should be presented in supporting information.

**Reply and Changes in the manuscript (2)** The procedures of mineral identification using EDXS, lattice fringe imaging, and electron diffraction are now provided in **supplement information** as attached at the end of this letter.

**Comment (3)** In the section 3.1, the relative abundance of clay mineral particles obtained from SEM-EDXS data (Table 1) for the Tenerife sample is directly compared with the total clay content (in wt%?) determined by XRD for a Cape Verde sample, but they are two different physical parameters. This point should be clarified.

**Reply (3)** Both data are incomplete because of the lack of either XRD or SEM-EDXS data. This is caused by the nonavailability of samples suitable for XRD in Tenerife and SEM-EDXS single particle analysis in Cape Verde (sampling campaigns were done many years ago).

**Changes in the manuscript (3)** We deleted the comparison of Tenerife SEM-EDXS data to Cape Verde XRD data.

**Comment (4)** In the section 3.3, the volume of iron (hydr)oxides in the dust particles is estimated. Two dimensions of the iron grains in dust are obtained from the bright-field TEM images, but how is measured the third dimension, notably for internal grains? From EDX maps? It should be explained.

**Reply (4)** The third dimension of iron (hydr)oxides was not measured. The percentage was derived by measuring the relative area of the iron (hydr)oxides from TEM images by counting pixel using the lasso tool and histogram of ADOBE PHOTOSHOP. In fact, the FIB slices cannot be used to measure the volume of mineral grains included in the dust particles because large portion of the included grains were commonly removed by the FIB work. In addition, we do not have any method to directly measure the volume of included submicron to nano-sized grains yet. However, the area% of included grains approximates the volume% (Vepraskas and Wilson, 2008, Soil Micromorphology: concepts, techniques, and applications, in: Methods of Soil Analysis, Part 5). Although the third dimension of iron (hydr)oxides could not be measured from the FIB slices, the area% of iron (hydr)oxide grains in a dust particle approximates their volume %. If the iron (hydr)oxide grains imaged by TEM are the only iron (hydr)oxides included in the dust particle, the volume of host particle would be enormously larger than the volume of tiny iron (hydr)oxides in comparison to corresponding areas because volume is proportional to the cube of the diameter. However, the FIB slices expose a cross section of iron (hydr)oxide grain which was randomly selected from many iron (hydr)oxide grains dispersed through the dust particle. Thus, area% approximates volume%. We would like to keep volume%.

**Changes in the manuscript (4)** We inserted following sentences: "Although the third dimension could not be measured, the area% of iron (hydr)oxide grains approximates volume% because tiny iron (hydr)oxide grains had been randomly dispersed through the dust particle (Vepraskas and Wilson, 2008)."

**Comment (5)** The section "3.5 Shape of dust particles" notably presents the methodology to measure the total particle volume, so it should be inserted before the part 3.3 in which particle volumes are presented and also discussed (sup Table 1).

**Reply (5)** We are interested in the volume of iron (hydr)oxides. The area of iron (hydr)oxides was measured by counting pixel using the lasso tool and histogram of ADOBE PHOTOSHOP. We did not measure their $a$, $b$, and $c$ dimensions.

**Changes in the manuscript (5)** We would like to keep part 3.5.

**Technical correction (6)** (Avila et al., 1997) is cited in the introduction, but (Avila et al., 1996) is listed in the bibliography.
**Reply and Changes in the manuscript (6)** The bibliography was corrected to 1997.

**Technical correction (7)** (Conny, 2013) is cited in the introduction, but it is not among the listed references.
**Reply and Changes in the manuscript (7)** Conny (2013) was added to the list.

**Technical correction (8)** Figure 1: What is the level high of the arrival air masses for the backward trajectory calculations by HYSPLIT? It should be mentioned in the Figure caption.
**Reply and Changes in the manuscript (8)** "Izana" was changed to Izana (2400 m asl.).

**Technical correction (9)** Figure 2: The meaning of yellow lines is given two times in the Figure caption.
**Reply and Changes in the manuscript (9)** One of the repeated sentences was deleted.

**Supplement Information – Mineral identification**

Minerals have their own crystal structures and chemical compositions. Thus, mineral identification using TEM is based on the lattice-fringe imaging and electron diffraction providing structural information and EDXS providing chemical information. Precise identification of all the minerals in the FIB slice to the species level is practically impossible because of beam damage, high vacuum, lower reliability of lattice fringes/electron diffraction data in comparison with XRD, and enormous time required.

**TEM identification of nonphyllosilicate minerals**

The identification of quartz, K-feldspar, plagioclase, calcite, amphibole, dolomite, titanite, apatite, and gypsum was straightforward based on their characteristic EDX spectra (Fig. S1). Although we could not identify mineral species of K-feldspar (sanidine, orthoclase, microcline), plagioclase (albite, oligoclase, andesine...), and amphibole (tremolite, actinolite, hornblende...) using time-consuming complex operation, the purpose of current mineral dust research was satisfied by grouping similar mineral species. Silica phase of the mineral dust from desert was almost quartz, consistent with XRD although few amorphous silica was identified by electron diffraction.

[Figure]

Fig. S1. TEM EDXS patterns of constituent minerals of dust particles obtained from the cross-sectional slices prepared by FIB.

**TEM identification of phyllosilicate minerals**

Phyllosilicate minerals were abundant in the mineral dust. The identification of muscovite, biotite, and chlorite was rather straightforward from their characteristic chemical compositions with the aid of lattice-fringe imaging (Fig. S2). However, the identification of nano-thin phyllosilicates (clay minerals) was difficult because of their breakdown under electron beam and small grain size below the minimum diameter of electron beam for EDXS. They occurred often as agglomerates. In addition, mixed layering of illite and smectite is common in natural soils. The identification of clay minerals was based on lattice fringes and chemical compositions: 1.0 nm for illite, ~1.0 nm for smectite and vermiculite, and ~7.0 nm for kaolinite (Fig. S2). Kaolinite was directly identified from its EDXS with the aid of lattice fringe imaging. However, illite, smectite, and illite-smectite mixed layers could not be separately identified each other because smectite was contracted under the high vacuum of the TEM chamber, showing ~1.0 nm lattice fringes similar to those of illite. Although EDXS can be used for identifying illite and smectite with interlayer cations K and Ca, respectively, they cannot be separately analyzed using EDXS, even when using an electron beam as small as possible. Therefore, we could not distinguish between nano-thin illite, smectite, and their mixed-layers, using conventional TEM work. To avoid over-interpretation, nano-thin platelets of clay minerals showing ~1.0 nm lattice fringes with varying K and Ca contents were grouped into illite-smectite series clay minerals (ISCMs). ISCMs are nano-scale mixtures of nano-thin platelets of illite, smectite, and illite-smectite mixed-layers.

Fig. S2. Identification of phyllosilicates using TEM-EDXS and lattice fringes.

**TEM identification of iron (hydr)oxides**

   Mineralogical identification of iron (hydr)oxides was also challenging. EDXS could not be used for the identification. Electron diffraction and lattice-fringe imaging should be used in combination as shown in Fig. S3. However, many iron (hydr)oxide grains could not be identified because of the overlap of many *d*-spacings, varying crystallographic orientation, and tiny grain sizes. Thus, we used species names only in cases in which mineral species were identified unambiguously by lattice fringe imaging and electron diffraction; in other cases, we used the collective term "iron (hydr)oxide".

[Figure]

Fig. S3. Identification of phyllosilicates using lattice fringes and electron diffraction.

**Mineralogical classification of dust particles using SEM-EDXS**

   Dust particles are essentially mixtures of mineral grains of diverse species and sizes. In case the quantity of powder dust samples is sufficient (~several hundred mg), XRD method is best for the determination of mineral composition. SEM-EDXS analyses of individual particle can be used when powder samples are insufficient or non-available. Ideally, mineral composition of individual dust particle can be determined by mixing several minerals to get the overall chemical composition of the particle. Then, the summation of the mineral compositions of thousands of dust particles considering their volume would lead to the mineral composition of bulk dust. However, the irregular morphology of dust particles prohibits the accurate determination of dust particles due to the difficulty of calibration. In addition, the chemical compositions of constituent minerals are varied. Prior to the development of reliable quantitative analysis procedure based on SEM for the mineral composition of individual dust particle, we adopted semi-quantitative approach. Since dust particles are generally dominated by one mineral species or group, we have determined the predominant mineral of a dust particle referring to the EDXS patterns of pure minerals as shown in Figs. S1 and S2. In case particles show intermediate EDXS pattern (Fig. S4), half of the particle was counted (0.5). Summation of the counts led to the approximate mineral composition of bulk dust. Although the procedure is evidently semi-quantitative, SEM-EDXS results were consistent with XRD results in the recent analyses of Asian dust (Table 1 in Park and Jeong (2016), Journal of the Mineralogical Society of Korea, 29, 79–87).

**SEM-EDXS**

[Figure]

Fig. S4. SEM-EDXS of dust particles.

Park and Jeong (2016)

**Table 1.** Mineral compositions of Asian dusts determined by XRD analysis and SEM-EDS single particle analysis

| Minerals | Asian dust XRD | | | | |
|---|---|---|---|---|---|
| | Feb 22 2015 | Mar 18 2014 | Mar 31 2012 | Mar 20 2010 | Average |
| | (This study) | (Jeong and Achterberg, 2014) | | | |
| ISCMs | 55 | 60 | 42 | 50 | 51 |
| Kaolinite | 2 | 1 | 3 | 4 | 3 |
| Chlorite | 5 | 3 | 6 | 7 | 5 |
| *Total clay* | *62* | *64* | *52* | *61* | 59 |
| Quartz | 18 | 14 | 23 | 15 | 17 |
| Plagioclase | 10 | 11 | 15 | 10 | 12 |
| K-feldspar | 4 | 0 | 6 | 2 | 3 |
| Amphibole | 0 | 0 | 1 | 2 | 1 |
| Calcite | 5 | 5 | 2 | 5 | 4 |
| Gypsum | 1 | 6 | 2 | 6 | 4 |
| Total | 100 | 100 | 100 | 100 | 100 |
| | SEM single particle analysis | | | | |
| ISCMs | 57 | 54 | 48 | 54 | 52 |
| Kaolinite | 2 | 1 | 3 | 2 | 2 |
| Chlorite | 3 | 2 | 4 | 6 | 4 |
| *Total clay* | *62* | *58* | *55* | *62* | 58 |
| Quartz | 19 | 19 | 21 | 17 | 19 |
| Plagioclase | 9 | 11 | 11 | 10 | 11 |
| K-feldspar | 3 | 4 | 5 | 3 | 4 |
| Amphibole | 0 | 1 | 1 | 0 | 1 |
| Calcite | 4 | 7 | 7 | 6 | 7 |
| Gypsum | 2 | 0 | 1 | 1 | 1 |
| Total | 100 | 100 | 100 | 100 | 100 |

We will prepare final version considering comment and reply above.

Sincerely

On behalf of co-authors

Gi Young Jeong
Corresponding Author